# The Chinese Spring Festival Impact on Air Quality in China: A Critical Review

**DOI:** 10.3390/ijerph19159074

**Published:** 2022-07-26

**Authors:** Guixian Wu, Wenling Tian, Li Zhang, Haiyan Yang

**Affiliations:** 1Nanchong Vocational and Technical College, Nanchong 637131, China; wuguixian1997@163.com (G.W.); lizh20222022@163.com (L.Z.); 2State Key Laboratory of Molecular Engineering of Polymers, Department of Macromolecular Science, Fudan University, Shanghai 200438, China

**Keywords:** air quality, PM_2.5_, Chinese Spring Festival, impact

## Abstract

It is known that the sharp change of air pollutants affects air quality. Chinese Spring Festival is the most important holiday for Chinese people, and the celebration of the holiday with fireworks and the movement of people all around the country results in significant change in multiple air pollutant emissions of various sources. As many cities and rural areas suffer from the air pollution caused by firework displays and more residential fuel consumption, there is an urgency to examine the impact of the Chinese Spring Festival on air quality. Hence, this paper firstly gives an overall insight into the holiday’s impact on ambient and household air quality in China, both in urban and rural areas. The main findings of this study are: (1) The firework displays affect the air quality of urban and rural atmosphere and household air; (2) the reduction in anthropogenic emissions improves the air quality during the Chinese Spring Festival; (3) the household air in urban areas was affected most by firework burning, while the household air in rural homes was affected most by fuel consumption; and (4) the short-term health impact of air pollution during the holidays also need more concern. Although there have been many publications focused on the holiday’s impact on ambient and household air quality, most of them focused on the measurement of pollutant concentration, while studies on the formation mechanism of air pollution, the influence of meteorological conditions, and the health outcome under the effect of the Chinese Spring Festival are rare. In the future, studies focused on these processes are welcomed.

## 1. Introduction

Air pollution has become a global public concern in recent years since more than 400 million deaths are associated with air pollution [1,2,3] and it also causes economic loss [4]. In China, the frequent haze events caught much attention of the public in recent decades [5,6,7]. Particulate matter (PM), specifically fine particles (PM_2.5_, particulate matter with diameter ≤ 2.5 μm), as one of the most concerning air pollutants, is the direct cause of haze and affects human health the most and climate change due to the toxic components such as organic carbon (OC), black carbon (BC), polycyclic aromatic hydrocarbons (PAHs), heavy metals, etc. [8,9,10,11]. Some gas pollutants such as methane (CH_4_), nitrogen oxide (NOx), and sulfur dioxide (SO_2_) also contribute to global warming and adverse health outcomes [12,13,14].

There is no doubt that emissions from human activities such as industry and vehicle emissions contribute significantly to poor air quality [15,16]. Numerous studies confirmed that a sharp change in air pollutant emission could significantly affect the local and regional air quality. For example, it was found that after the lockdown due to COVID-19, the ambient air quality improved considerably. During the stringent COVID-19 lockdown, less traffic and industrial emissions resulted in notable ambient air pollution reduction in many countries. In India [17], the COVID-19 lockdown caused a considerable air quality improvement. It is reported that the concentrations of PM_10_ and PM_2.5_ witnessed the greatest reduction (>50%) in comparison to the pre-lockdown period. NO_2_ (−52.68%) and CO (−30.35%) levels also significantly reduced during the lockdown period compared to the pre-lockdown period. Similar results were found in many other cities or countries, such as the city of Salé in Morocco, Wuhan in China, Rome in Italy, and the U.S. [18,19,20]. The explosive publications focused on the impact of COVID-19 all confirmed that the sharp reduction in various sources of air pollutants could significantly affect the atmosphere quality and an expected improvement could be achieved. Meanwhile, limited studies realized that other special periods might also have similar results when a sharp change in air pollutants emissions occurs such as the special holidays in China, including the “Chinese Spring Festival”.

Chinese Spring Festival is an annual festival in the last thousand years in China, which is the most important holiday for Chinese people. It was well believed that the emissions of air pollutants during Chinese Spring Festival result from a very sharp change due to the special culture in China during this period, which is: (1) young adults working in big cities always travel back to their hometown to celebrate the holidays with their families [21]; (2) fireworks in many areas are still permitted while in some cities emissions are now forbidden [22]. The largest migration of the population all over the world resulted in various situations in different cities or rural areas. In some megacities such as Beijing and Shanghai, the migration of immigrants resulted in lower industry and vehicle emissions, while in some small cities and rural areas, the people traveling from megacities caused an increase in vehicle emissions and household air pollutant emissions associated with residential energy use. These different situations vary the impact of Chinese Spring Festival on air quality. However, in recent years, many researchers have focused on the air quality during this special period. To date, there is no critical review that fully investigates the impact of Chinese Spring Festival on air quality, not only on ambient air, but also on household air. In this paper, both English and Chinese studies were selected and reviewed to have a full insight into the impact of annual holidays on air quality and prioritized studies in the future are suggested. It is expected that this review will bring new knowledge on the impact of sharp emission changes in air pollutants on the air quality during annual holidays.

## 2. Materials and Methods

The literature search was conducted in both English and Chinese databases. The search of peer-reviewed papers in English was carried out with the database of Web of Science, which well covered all important literature. Meanwhile, the database of China National Knowledge Infrastructure (CNKI) was chosen for the selection of papers in Chinese. The search terms used in this study include “air”, “air pollution”, “PM_2.5_”, “CO”, “NOx”, etc. AND “Chinese Spring Festival”. There are 294 and 257 results in English and Chinese in the databases, respectively. Studies focused on the Chinese Spring Festival impact on air pollution started in 2006, and the publication numbers rose sharply from 2018, which indicated more and more attention was paid to this topic. As shown in Figure 1, Jiangsu Province (22%) is the place where the most measurements were conducted, followed by Beijing (11%), however, studies in some provinces such as Tibet, Qinghai, Ningxia, and Macao are rare, as each of them represents only 1% of the total provinces. Then, the selected papers were screened according to the title and abstract. The papers covering only a single Spring Festival were not included since there are no comparisons of the air quality between the holidays and that before and after the holidays or that during the holidays in other years available. Only the studies carried out during and before/after the holidays or across the holidays of several years were chosen for further review.

## 3. Results

### 3.1. The Impact on the Ambient Air Pollution

#### 3.1.1. Ambient Air in Urban Areas

Most of the available studies focused on the impact of the Chinese Spring Festival on the ambient air in urban areas based on the measurement of air pollutants including gas pollutants and/or components bound to PMs. In Beijing, the capital of China, it was found that, except for the ozone (O_3_) which remained stable over recent years, other air pollutants such as PM_2.5_, PM_10_, NO_2_, SO_2_, and CO decreased in the Spring Festival of 2018 and 2019 compared with the previous years from 2014 to 2017. The result confirmed that the strict firework display control since 2018 contributed most to the air quality improvement during the holidays [23]. As the capital of China, Beijing has a large migrant population, which means the people living in Beijing during the annual holidays will sharply decline. It was found that the tropospheric NO_2_ column density decreased by 41.6% and rebounded by 22.3% after the Chinese Spring Festival in Beijing from 2013 to 2018, mainly caused by the reduction in traffic intensity [24]. The drop in air pollution during the special period is called “the Spring Festival effect”, which is also supported by a study conducted in Jinan, the capital city of Shandong Province, northern China. It was found that the water-soluble secondary organic aerosol (WSOA) concentrations decreased during the holidays compared with those not during the Spring Festival [25]. Additionally, in Xiamen, the capital city of Fujian Province, southern China, it was reported that due to the reduction in the air pollutant emissions from factory production, construction, and vehicles during the Spring Festival, the concentrations of organic carbon, elemental carbon, and water-soluble ions in PM_2.5_ decreased by 79%, 84%, and 28%, respectively, compared with those before the Spring Festival. Interestingly, the concentrations of metal elements such as K^+^, Mg^2+^, Al, Sr, and Ba increased by 3122%, 572%, 184%, 180%, and 138%, respectively, which was caused by the display of fireworks and firecrackers during the Spring Festival [22]. In addition, in places such as India, studies have also found that the display of fireworks and firecrackers during firework festivals led to a rapid increase in PM_2.5_ [26,27,28]. Additionally, the research carried out in the city of Debrecen (Hungary), higher concentrations of Ca, Mg, and Sr were detected in leaf samples collected during a festival [29]. Furthermore, Alexandre et al. [30] reported that the levels of PM_2.5_ and elements such as K, Cl, Al, Mg, and Ti were markedly higher in plume-exposed filters during the nine launches of the 2007 Montreal International Fireworks Competition. This indicates that the impact of fireworks on air pollution is a worldwide problem that requires attention. It should be noted that although the primary emissions of air pollutants in megacities might decrease, the secondary aerosol formation might unexpectedly increase, which indicates a strong need for O_3_ control [31].

It was easy to find that the above studies were mostly conducted in cities that had restricted firework display control. In some cities, firework burning was not forbidden, which caused a different festival effect from the megacities mentioned above. For example, in the city of Lanzhou, the capital of Gansu Province, northern China, it was found that firework displays contributed 37.0% to the number concentration of particles less than 1000 nm, which was 2–6 times higher than the period before the holidays [32]. A study in Changchun, the capital city of Jilin Province, northern China, found that the burning of fireworks could significantly contribute to the increase in heavy metals in PM_1.0_, which could result in non-negligible health impacts on local residents with an excess cancer risk beyond the safety limit (10^−6^) [33]. Using Fourier transform ion cyclotron resonance mass spectrometry, molecular characterization of firework-related urban aerosols was carried out and determined to explore the effect of fireworks during the Chinese Spring Festival, and it was found that the number concentration of sulfur-containing compounds increased dramatically associated with the firework display [34]. Similarly, it has been reported that PM levels increase significantly during firework festival displays all over the world, and the high PM concentrations remain suspended in the air for as long as one month [35]. For example, Camilleri et al. [36] reported that PM10 and metal concentration levels were significantly higher during the firework festival in Malta and Gozo. In addition, Khaparde et al. [37] reported that PM10 levels were increased by four to nine times and associated barium (Ba) levels were increased by eight to twenty times in India during fireworks. Additionally, it was reported that aerosol including NO and SO_2_ was relatively higher than normal during fireworks of Las Fallas in Valencia [38]. Therefore, it was easy to confirm that if cities did not have firework displays, the air pollution would decline, while it would be increased when fireworks are burnt during the holidays.

#### 3.1.2. Ambient Air in Rural Areas

Studies also focused on the ambient air quality affected by the Chinese Spring Festival in rural areas. A one-month measurement of BC conducted in a rural site located in Henan Province, northern China, found that the BC concentrations during the holidays were higher than in the periods before and after the holidays. The contribution of biomass burning in local and surrounding provinces increased during the Chinese Spring Festival, which confirmed that the biomass combustion in rural areas during the holidays was crucial for the ambient air pollution [39]. It proved that the change in the emission sources during the Chinese Spring Festival could affect the air quality. Xie et al. [40] found that during the Chinese Spring Festival, most emission sources of volatile organic compounds (VOCs) were sharply reduced, thus the VOC concentration in a rural site located in northern China declined by 49% compared with the period before the holiday. By utilizing a machine learning technique, Hu et al. [14] found that the ammonia (NH_3_) concentration decreased during the Spring Festival compared with the period before and after the holidays in a rural area in northern China, which was caused by lower anthropogenic emissions, although this reduction is weaker than that in urban areas. Interestingly, Zhao et al. [41] found that in the city of Hefei, Anhui Province, southern China, the firework burning in rural areas could firstly give rise to PM_2.5_ and related metal elements, then affect the air quality in urban Hefei, where firework burning was banned. This study proved that to achieve a good air quality during the Spring Festival, a firework ban only in urban areas might not be enough. Thus, a firework ban both in rural and urban areas is needed. This result is supported by a study conducted in “2 + 26” cities (the Chinese term that refers to the air pollution transmission channel in Beijing, Tianjin, Shandong, and Hebei) in northern China, which found an effective reduction in PM_2.5_ and other air pollutants due to firework prohibition and restrictions in the rural and urban areas, while firework burning in the surrounding rural area still worsened the air quality in some urban areas [42].

### 3.2. The Impact on Household Air Pollution

Compared with the studies on ambient air quality, there are limited studies focused on the impact of Chinese Spring Festival on household air pollution. As shown in Figure 2, firework burning and indoor cooking during the Chinese Spring Festival will greatly impact the household air pollution. Qi et al. [43] and Du et al. [11] measured the household PM_2.5_ before, during, and after Chinese Spring Festival in urban and rural households using real-time PM_2.5_ monitors, respectively. It was found that in Nanchong, a middle city located in Sichuan Province, southern China, which has not yet implemented a strict ban on firework displays, firework burning significantly affected the ambient and indoor air quality. It was found that very high peaks in the afternoon or at midnight of the Spring Festival were observed. During the afternoon of New Year’s Eve, the ambient and indoor PM_2.5_ gradually rose from 50 μg/m^3^ to 200 μg/m^3^ in a few hours. At 0:00 of the first day of the Chinese Spring Festival, more fireworks were burnt, the ambient PM_2.5_ quickly reached a peak of more than 300 μg/m^3^, and the indoor PM_2.5_ also increased to about 250 μg/m^3^. Five or six days later, the ambient and indoor PM_2.5_ decreased to pre-New Year levels. The results confirmed the significant effect of firework burning on ambient and indoor PM_2.5_ using high time resolution data, and also supported the rationality of firework bans in many big cities during the holiday. A similar study in an urban household located in Zhengzhou, Henan Province also confirmed that the firework burning resulted in a high peak of ambient PM_2.5_ and then affected the indoor PM_2.5_ through outdoor–indoor air exchange [44]. In rural homes of Hunan Province, southern China, a sharp increase in kitchen and living room PM_2.5_ concentration during the Chinese Spring Festival was found compared with that before the holidays, while the ambient PM_2.5_ decreased. The rise in indoor PM_2.5_ was caused by more solid fuels needed for cooking and space heating since more people were living in the rural homes during the holidays, and the reduction in industry and vehicle emission caused the decrease in the ambient PM_2.5_ [45]. The more severe household air pollution during the Spring Festival was found to affect blood pressure, increasing from 131 ± 18 mmHg (before the Chinese Spring Festival) to 135 ± 22 mmHg (after the Chinese Spring Festival) of the rural elders, though the blood pressure was not obviously affected, and Du et al. [46] reported that the rural residents with higher blood pressure suffered from a significant increase in SBP during the Chinese Spring Festival, which indicated the health impact of higher household air pollution under the effect of the holidays. This result is also supported by another study conducted in rural homes located in Hebei Province, northern China, which found potential increases in solid fuel consumption since more people spending more time indoors could result in increased household air pollution and impact the health of rural residents [47]. The above results also showed the benefit of clean energy used in rural areas since the households using gas fuel and electricity usually had lower indoor air pollution while under the influence of the Chinese Spring Festival.

In the above studies, real-time PM_2.5_ monitors make it possible to explore the dynamic change in PM_2.5_, which provided some valid and new insight into the holiday impact. It is well known that the real-time PM_2.5_ monitors could collect high-resolution data, but they can only measure a limited number of pollutants, such as PM_2.5_, CO, CO_2_, SO_2_, etc., and they are incapable of collecting the component information of household PM_2.5_, such as PAH, heavy metals, etc. The component information of household PM_2.5_ was also scarce. A study was conducted in four indoor environments such as office and laboratory, and an outdoor environment in urban Harbin, Heilongjiang Province, northern China. PM_2.5_ was sampled and determined for the PAHs before and during the Spring Festival, which found the indoor PAH concentration during the Spring Festival was lower than that before the holidays [48], in contrast to the result of the outdoor environment [49,50]. It was reported that the highest concentrations of total PAHs (138 ± 1.70 and 109 ± 8.80 ng/m^3^) were found in outdoor concentrations before and during the Spring Festival, respectively, while the lowest total concentrations (84.9 ± 4.90 and 63.0 ± 14.0 ng/m^3^) were found in the indoor concentrations. Both before and during the Spring Festival, the PM_2.5_-bound PAH concentrations in the outdoor environment were related to the indoor environment [48]. This is easy to understand since during the holidays, people no longer stayed in these selected indoor environments and the doors and windows were always closed, which limited the outdoor–indoor air exchange. At this stage, it is confirmed that PAH concentrations under the impact of the Chinese Spring Festival are very low, especially in rural homes, which is worthy of investigating in future studies.

### 3.3. Limitations and Implications

As a typical period of significant change in emission sources of air pollutants, the Chinese Spring Festival is an ideal period for research on air pollution including the pollution mechanism, source pattern difference, etc. In addition, the air pollution in this special period is very crucial for the estimation of short-term impact on human health. Previous studies mostly focused on the measurement of air pollutants, through which different effects of the Chinese Spring Festival with/without firework displays in both cities and rural areas was discussed. As the impact of household air pollution has become more clear in recent years, the effect of the holidays on household air pollution both in rural and urban areas was also explored. Compared with the studies that reported the concentrations of air pollutants including gas pollutants and components bound to PMs, the source, formation mechanism of the air pollution during the special period is relatively rare. Through this review, some limitations are found and need to be addressed in the future. First, many studies focused on the firework burning, while limited studies tried to estimate the influence of other factors such as the industry and vehicle emission change and meteorological conditions. Second, the studies on the air in rural areas and households are also limited, resulting in some knowledge gaps on the festival impact on household air to date. Third, more information about PM, such as concentration, size, morphology, etc., should be obtained to ascertain the influence of PM during the Chinese Spring Festival. Finally, the health impact associated with the air pollution during this special period is still not well understood. Future work is urgently needed, which should not only focus on the air pollutant concentration measurement, but also the formation mechanism of air pollution, the influence of meteorological conditions, and the health outcomes.

## 4. Prioritized Study Suggestions

As mentioned above, there are some limitations of current studies at this stage, which limited the accurate understanding of the holiday impact on air quality in China. Hence, some suggestions for prioritized studies are given in detail below:

(1) Well-designed studies are strongly recommended. It would be better to measure the air quality and/or sample before, during, and after the holidays and the resolution should be as high as possible. The sampling duration could be hours, or even based on real-time monitors. Meteorological data including wind direction, relative humility, and temperature should be measured.

(2) There are limited studies on the household air quality affected by the Chinese Spring Festival, both in rural and urban households, especially considering the large spatial variation in China. Studies in different areas and more households are urgently needed. Given there are some studies using real-time PM_2.5_ monitors to explore this issue, additional studies using filters are also welcomed to investigate the effect of the holidays on the components of household PM_2.5_.

(3) Some more detailed information about PM, such as sizes, morphology, ultrafine particles, various PM chemical components, and elevated levels, should be measured.

(4) It would be interesting to explore the health impact of the holidays due to the variations in air pollutant exposure. The highly different levels of exposure to air pollution during the Chinese Spring Festival might cause short-term health impacts, especially on some sensitive people such as elders, children, and pregnant women.

## 5. Conclusions

This paper conducts a critical review on the impact of Chinese Spring Festival on the ambient and household air quality in urban and rural areas. The result shows that the cities with firework displays experience a rise in air pollutants during the holidays, while in the cities without firework burning, the air pollution declines due to lower industry and vehicle emissions since many migrant populations leave the cities during this period for their hometown. As rural areas are usually without any ban on fireworks, the air quality in rural areas is much worse during the festival compared with the period before and after the holidays, and then the pollution associated with fireworks affects the ambient air of the surrounding cities. Household air in urban areas was mostly affected by firework burning, while the air pollutants in rural areas were discharged due to the greater solid fuel consumption. This paper confirmed that the firework bans, both in rural and urban areas, benefit air quality, and also shows the benefit of clean energy on household air of rural homes.

## Figures and Tables

**Figure 1 ijerph-19-09074-f001:**
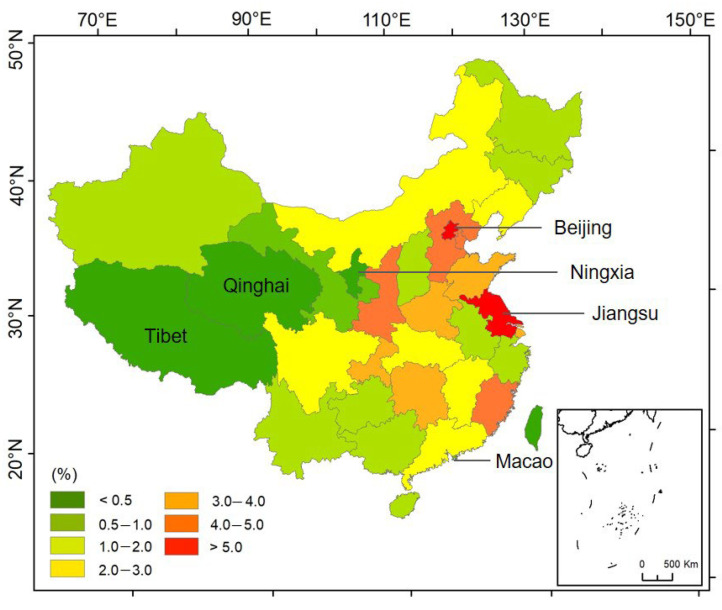
The percentages of studies focused on the Chinese Spring Festival by region.

**Figure 2 ijerph-19-09074-f002:**
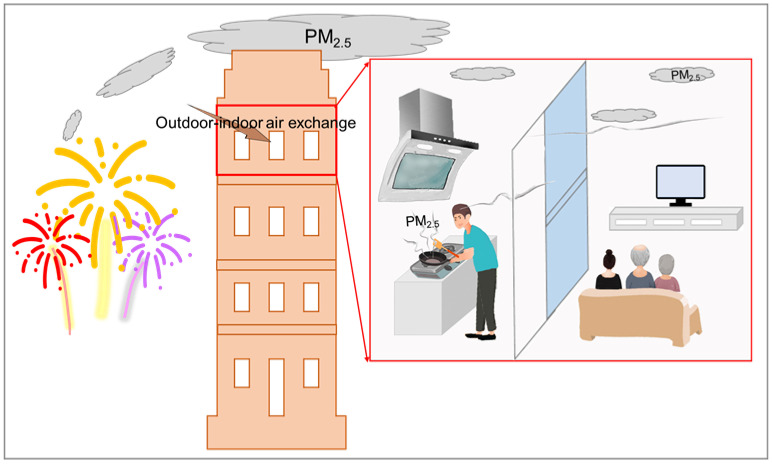
The household air pollution under the impact of firework burning and indoor cooking during the Chinese Spring Festival.

## Data Availability

Not applicable.

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
