# Peer review of "The Chinese Spring Festival Impact on Air Quality in China: A Critical Review"

_ijerph, 2022, doi:10.3390/ijerph19159074_

Round 1

Reviewer 1 Report

See Attached.  In general, the results are clear but far from compelling.   To add interest to this paper, add a compelling plot, and make stronger cases for the health effects and the necessity for more study. 

General Comments:
Abstract: You need to create a sense of urgency here. “Could” and “non-negligible” are weak
descriptors. Why is it essential to study air quality during this period?

It would be useful to include more information from the studies that examined the health
effects of short-term exposure to high-levels of pollution. You mention changes to blood
pressure (by how much). What else?

To add urgency, perhaps add a plot summarizing the impact of the Festival on ambient and
indoor air quality as determined by the literature search.

I would also use the abstract to make the case for additional studies. It is too general as
written.

You mention 294 English and 257 Chinese papers but only cite 39 papers. Some statistics on
these papers would be useful. How many focused on emissions, meteorology, air quality,
health, etc. These statistics would be useful for making the case that more study is needed in
certain areas. Presumably you limited the study to 39 papers because they covered multiple
Festivals or the period before, during, and after the Festival.

L237: Expand on the components of a “well designed study”. What resolution is needed? What
meteorological data should be sampled. Is there a role for low cost monitors in this study?

Minor Comments:

L22: Quantify what you mean by non-negligible

L104: Are the 41.6% and 22.3% values means for one Festival or multiple Festival?

L146-148: Why were the VOC emissions reduced?

L158: Explain what “2+26” cities are for the non-Chinese reader

L116: Too many “significant” digits for the % values ... Perhaps go with 3120%, 570%, etc.

L163-194: What is the relative importance of outdoor-indoor exchange versus increases in
cooking and space heating on indoor air quality during the Festival? Is there a study that
examines this?

L202: The very low outdoor-indoor exchange in this study seems to contrast with study [35]
cited in line 180. Please reconcile the differences.

L210: Add more text to caption.

Grammatical Comments:

Could is a weak verb to use. It indicates that there is little certainty as to whether air pollutant
changes affect air quality or health. Best to use a stronger verb.

L10: It was already known It is known

L10: could affect affects

L12: migration population migration of people

L17: display could significantly display affects
L19: could improve improves

L22: Although there had been Although there have been

L23: while most of them focused most of them are focused

L30: globally public global public

L40: It was no There is no

L41: contributed contribute

L61: the youngers working young adults working

L63: fireworks in many areas is fireworks in many areas are

L64: it is forbidden they are forbidden

L79-80: ... Chinese databases given the ... Chinese databases.

L81: Web of science Web of Science

L84: in this study was as pollutants (air ... ) in this study include air, ... etc.

L88: between during the between the

L96: urban area urban areas

L100: restrict strict

L107: proved by supported by

L119: might unexpected increase might unexpectedly increase

L121: had restrict firework display control had restricted firework displays

L141: were the highest when compared with the period before were higher than the periods
before

L163: researches on ambient air studies on ambient air quality

L179: high peak of ambient PM2.5 increase high peak of ambient PM2.5

L181: China, it was found that sharp China, a sharp

L182: Festival compared Festival was found compared

L186: severer more severe

L191: finally treated impact

L198-203: This sentence is too long and convoluted. Break into 2 or 3 sentences.

L206: components of household PM2.5 that PAH concentrations under

L207: very scare very scarce

L214: In the other hand, In addition,

L218-219: As the household air pollution is more and more concerned in the last years As
the impact of household air pollution became more clear in recent years

L222: formation mechanism information on the formation mechanism

L223-224: and welcomed to be addressed in the and need to be addressed in the

L240: limited researches on - limited studies on

L253: could obtain obtain

L254: might decline declines

L257: much worse compared much worse during the Festival compared

Author Response

General Comments:

Comment 1: Abstract: You need to create a sense of urgency here. “Could” and “non-negligible” are weak descriptors. Why is it essential to study air quality during this period?

Response 1: Thank you for the constructure advise, we have revised the Abstract in line 10-28: “It is known that the sharp change of air pollutants affects air quality. Chinese Spring Festival is the most important holiday for Chinese people, and the celebration of the holiday by firework and the migration of people all around the country resulted in significant change of multiple air pollutants emissions of various sources. As many cities and rural areas were suffering from the air pollution caused by fireworks display and more residential fuel consumption, there is an urgency to catch more concern on the impact of the Chinese Spring Festival on air quality. Hence, this paper firstly had a whole insight of the holiday impact on ambient and household air quality in China, both in urban and rural areas. The main findings of this study are: (1) The firework display affects the air quality in urban and rural atmosphere and household air; (2) the reduction of anthropogenic emissions improves the air quality during the Chinese Spring Festival; (3) the household air in urban and rural homes was affected most by firework burning and more solid fuel consumption, respectively; and (4) the short-term health impact of air pollution during the holidays also need more concern. Although there had been many publications focused on the holiday impact on ambient and household air quality, most of them focused on the measurement of pollutants concentration, studies on the formation mechanism of air pollution, the influence of meteorological condition, and the health outcome under the effect of the Chinese Spring Festival are rare. In the future, studies focused on the household air pollution and the associated health outcome under the impact of the Chinese Spring Festival, the formation mechanism of air pollution, the influence of meteorological condition are welcomed.”

Comment 2: It would be useful to include more information from the studies that examined the health effects of short-term exposure to high-levels of pollution. You mention changes to blood

pressure (by how much). What else?

Response 2: Thank you for the constructure advise, we have added the detail information about the changes to blood pressure in line 260-263: “The more severe household air pollution during the Spring Festival was found to affect the blood pressure from 131±18 mmHg (before the Chinese Spring Festival) to 135 ± 22mmHg (after the Chinese Spring Festival) of the rural elders, which confirmed the short-term health impact of higher household air pollution under the effect of the holidays”

Comment 3: To add urgency, perhaps add a plot summarizing the impact of the Festival on ambient and indoor air quality as determined by the literature search.

Response 3: Thank you for the constructure advise, we added the sentence in line 13-15: “As many cities and rural areas were suffering from the air pollution caused by fireworks display and more residential fuel consumption, there is an urgency to catch more concern on the impact of the Chinese Spring Festival on air quality.”

Comment 4: I would also use the abstract to make the case for additional studies. It is too general as written.

Response 4: Thank you for the constructure advise, we revised the sentence as in line 22-28: “Although there had been many publications focused on the holiday impact on ambient and household air quality, most of them focused on the measurement of pollutants concentration, studies on the formation mechanism of air pollution, the influence of meteorological condition, and the health outcome under the effect of the Chinese Spring Festival are rare. In the future, studies focused on the household air pollution and the associated health outcome under the impact of the Chinese Spring Festival, the formation mechanism of air pollution, the influence of meteorological condition are welcomed.”

Comment 5: You mention 294 English and 257 Chinese papers but only cite 39 papers. Some statistics on these papers would be useful. How many focused on emissions, meteorology, air quality, health, etc. These statistics would be useful for making the case that more study is needed in certain areas. Presumably you limited the study to 39 papers because they covered multiple Festivals or the period before, during, and after the Festival.

Response 5: Thank you for the constructure advise, we added an Analysis of the selected papers was shown in Figure 1. And some new information was added accordingly in line 113-118: “Studies focused on the Chinese Spring Festival impact on air pollution started in 2006, and the publications numbers raised sharply from 2018, which indicated more and more concern was paid on this topic. As shown in Figure 1, Jiangsu province is the place where the most measurements were conducted, following by Beijing, however, studies in some provinces such as Tibet, Qinghai, Ningxia, and Macao are really rare.”

We want to state why we only cite limited publications. The main reason is most of the selected papers simply done the similar studies in different cities, thus, we only cite publications with high quality.

Comment 6: L237: Expand on the components of a “well designed study”. What resolution is needed? What meteorological data should be sampled. Is there a role for low cost monitors in this study?

Response 6: Thank you for the constructure advise, some information is added in line 325-337: “The sampling duration could be hours, or even based on real-time monitors. Meteorological data including wind direction, relative humility, and temperature should be measured.”

Minor Comments:

Comment 7: L22: Quantify what you mean by non-negligible

Response 7: Thank you for the constructure advise, we revised it as in line 21-22: “ the short-term health impact of air pollution during the holidays also need more concern.”

Comment 8: L104: Are the 41.6% and 22.3% values means for one Festival or multiple Festival?

Response 8: Yes, here it means multiple Festival. We revised the manuscript in line 147-151: “As the capital of China, Beijing has a large quantity of migrant population, which means the people lived in Beijing during the annual holidays will sharply decline. It was found the tropospheric NO2 column density decreased by 41.6% and rebounded by 22.3% after the Chinese Spring Festival in Beijing counted from 2013 to 2018, mainly caused by the reduction of traffic intensity”

Comment 9: L146-148: Why were the VOC emissions reduced?

Response 9: As the capital of China, Beijing has a large quantity of migrant population, which means the people lived in Beijing during the annual holidays will sharply decline. The traffic and industry emission will reduce accordingly.

Comment 10: L158: Explain what “2+26” cities are for the non-Chinese reader

Response 10: we explain the “2+26”cities in line 220-225: “This result is supported by a study conducted in “2+26” cities (the Chinese term that refers to the city of air pollution transmission channel in Beijng, Tianjin, Shandong, and Hebei) in northern China, which found an effective reduction in PM2.5 and other air pollutants due to the fireworks prohibition and restrictions in the rural and urban areas, while the fireworks burning in the surrounding rural area still worsen the air quality in some urban areas”.

Comment 11: L116: Too many “significant” digits for the % values … Perhaps go with 3120%, 570%, etc.

Response 11: we revised the % values in line 158-162: “the concentrations of organic carbon, elemental carbon and water soluble ions in PM2.5 decreased by 79%, 84% and 28%, respectively, compared with those before the Spring Festival. Interestingly, the concentrations of metal elements such as K+, Mg2+, Al, Sr, and Ba increased by 3122%, 572%, 184%, 180%, and 138%, respectively, which was caused by the display of fireworks and firecrackers during the Spring Festival”.

Comment 12: L163-194: What is the relative importance of outdoor-indoor exchange versus increases in cooking and space heating on indoor air quality during the Festival? Is there a study that

examines this?

Response 12: Sorry, none studies focused on this topic. We think this topic is worth to study in the future.

Comment 13: L202: The very low outdoor-indoor exchange in this study seems to contrast with study [35] cited in line 180. Please reconcile the differences.

Response 13: Yes, we mentioned in the paper the result in the original paper. The contrast result in Heilongjiang because during the holidays, people no longer stayed in these selected indoor environments and the doors and windows were always closed, which limited the outdoor-indoor air exchange.

Comment 14: L210: Add more text to caption.

Response 14: Thank you for the constructure advise, the caption of Figure 2 was revised as in line 293-294: “The household air pollution under the impact of fireworks burning and indoor cooking during the Chinese Spring Festival.”

Grammatical Comments

Could is a weak verb to use. It indicates that there is little certainty as to whether air pollutant

changes affect air quality or health. Best to use a stronger verb.

L10: It was already known → It is known

L10: could affect → affects

L17: display could significantly → display affects

L19: could improve → improves

L22: Although there had been → Although there have been

L23: while most of them focused → most of them are focused

L30: globally public → global public

L40: It was no → There is no

L41: contributed → contribute

L61: the youngers working → young adults working

L63: fireworks in many areas is → fireworks in many areas are

L64: it is forbidden → they are forbidden

L79-80: … Chinese databases given the … → Chinese databases.

L81: Web of science → Web of Science

L84: in this study was as pollutants (air … ) → in this study include air, … etc.

L88: between during the → between the

L96: urban area → urban areas

L100: restrict → strict

L107: proved by → supported by

L119: might unexpected increase → might unexpectedly increase

L121: had restrict firework display control → had restricted firework displays

L141: were the highest when compared with the period before → were higher than the periods

before

L163: researches on ambient air → studies on ambient air quality

L179: high peak of ambient PM2.5 increase → high peak of ambient PM2.5

L181: China, it was found that sharp → China, a sharp

L182: Festival compared → Festival was found compared

L186: severer → more severe

L191: finally treated → impact

L198-203: This sentence is too long and convoluted. Break into 2 or 3 sentences.

L206: components of household PM2.5 → that PAH concentrations under

L207: very scare → very scarce

L214: In the other hand, → In addition,

L218-219: As the household air pollution is more and more concerned in the last years → As

the impact of household air pollution became more clear in recent years

L222: formation mechanism → information on the formation mechanism

L223-224: and welcomed to be addressed in the → and need to be addressed in the

L240: limited researches on -→ limited studies on

L253: could obtain → obtain

L254: might decline → declines

L257: much worse compared → much worse during the Festival compared

Response: Thanks very much for your nice comments. Revised accordingly.

Reviewer 2 Report

In my opinion, it is a very interesting paper. It is very well structured and organized and the information provided as well as the suggestions are very important to take into account in future research and also for the authorities with competences in matters of air quality.

I only have one comment and that is that please review the instructions for authors since the references are not all uniform: in some the name of the journal goes with the first letters in capital letters, in others it goes with the abbreviation. In the text, for example in lines 37, 149, 164, there is a comma before the bracket of the references.

Author Response

General Comments:

In my opinion, it is a very interesting paper. It is very well structured and organized and the information provided as well as the suggestions are very important to take into account in future research and also for the authorities with competences in matters of air quality.

I only have one comment and that is that please review the instructions for authors since the references are not all uniform: in some the name of the journal goes with the first letters in capital letters, in others it goes with the abbreviation. In the text, for example in lines 37, 149, 164, there is a comma before the bracket of the references.

Response:Thanks for your comments. We all feel encouraged due to your positive comments. We revised the manuscript accordingly.

Reviewer 3 Report

A brief summary: 

Guixian et al. present the Chinese spring festival impact on air quality in China: A critical review, which is very interesting, and I think with extra work, it can be substantially improved, becoming a valuable contribution to the literature.

At this stage, the manuscript can not be published; a future investigation is needed.

Unfortunately, the scope of work and obtained results are not sufficiently detailed and do not provide novel and innovative conclusions for future research. Generally, this manuscript requires correction, and more information should be added.

General concept comments:

The manuscript is focused on a literature review of a single annual event in China. This approach is not sufficient to argue for a review paper.

I strongly recommend adding more information about fireworks’ impact at different sites ( across the world).  Even though the study is focused on China, a subsection that should include the results from fireworks in other regions can be an added value to the paper.

The presented results from previous research are not sufficient argued and comparatively treated. More comparative results should be extensively treated to evaluate the actual research stage’s pros and cons. I recommend that the authors be more specific and exclude affirmations like “some studies” without a clear reference and traceability.

Analysing different regions in China, a map to show the research distribution will be useful for non-Chinese readers.

 The suggestions made by the authors are not quantifiable and do not represent an added value to the actual research stage on this topic.

This review should represent an added value for the research made until now (e.g. https://doi.org/10.1080/10962247.2016.1219280)

Specific comments:

Will be added after reconsideration of the paperwork

Author Response

A brief summary:

Guixian et al. present the Chinese spring festival impact on air quality in China: A critical review, which is very interesting, and I think with extra work, it can be substantially improved, becoming a valuable contribution to the literature.

At this stage, the manuscript can not be published; a future investigation is needed.

Unfortunately, the scope of work and obtained results are not sufficiently detailed and do not provide novel and innovative conclusions for future research. Generally, this manuscript requires correction, and more information should be added.

Response: Thanks for your comments. Following all the reviewers’ suggestions, the paper is revised, we hope the new version could address all your concern.

General concept comments

The manuscript is focused on a literature review of a single annual event in China. This approach is not sufficient to argue for a review paper.

Comment 1: I strongly recommend adding more information about fireworks’ impact at different sites ( across the world).  Even though the study is focused on China, a subsection that should include the results from fireworks in other regions can be an added value to the paper.

Response 1: Thank you for the constructive advise, we Added accordingly in line 195-197: “Similarly, it has reported that PM levels increase significantly during the firework displays all over the world, and the high PM concentrations remains suspended in the air as long as one month [32]”.

Comment 2: The presented results from previous research are not sufficient argued and comparatively treated. More comparative results should be extensively treated to evaluate the actual research stage’s pros and cons. I recommend that the authors be more specific and exclude affirmations like “some studies” without a clear reference and traceability.

Response 2: Thanks for your advice. We revised accordingly.

Comment 3: Analysing different regions in China, a map to show the research distribution will be useful for non-Chinese readers.

Response 3: Thank you for the constructive advise, the publications on the topic of this review is analyzed and shown in Figure 1.

Comment 4: The suggestions made by the authors are not quantifiable and do not represent an added value to the actual research stage on this topic.

Response 4: Thank you for the constructive advise, we revised the suggestions in line 344-345: “Some more detail information such as sizes, morphology, ultrafine particles, various PM chemical components, and elevated levels about PM should be measured.”

Comment 5: This review should represent an added value for the research made until now (e.g. https://doi.org/10.1080/10962247.2016.1219280)

Response 5: This paper is very valuable, cited accordingly and some useful information is compared with our paper.

1.We added the sentence in line 163-179: “Besides, in the other place such as India, studies have also found that the display of fire-works and firecrackers during the firework Festivals lead to a rapid increase in PM2.5 [26,27]. This indicates that the impact of fireworks on air pollution is a worldwide problem that needs to be draw attention.”

2.We added the sentence in line 312-314: “Third, more information about PM such as concentration, size, morphology, etc., should be counted to reflex the influence of PM during the Chinese Spring Festival.”

Specific comments:

Will be added after reconsideration of the paperwork

Response: We hope to see your specific comments soon.

Round 2

Reviewer 1 Report

Main Comments:

The grammar is still sub-standard.  I’ve suggested some changes.  More could be made. 

I’m not certain what you mean by “respectively” on L23 and L296.  Does this mean fireworks are more important than solid fuel consumption? If yes, please say this more clearly.  To put it another way, can you answer this question using the literature? During the festival, which is more important for indoor air quality 1) outdoor to indoor exchange of PM from fireworks or 2) more solid fuel consumption within houses.  

Figure 1: Please label provinces discussed in the text (Jiangsu, Beijing, Tibet, Qinghai, Ningxia, and Macao)

L97-98: How can one tell from the plot that there are more publications from Jiangsu than Beijing? 

L213-214:  The difference between 131 (before) and 135 (small) mmHg is small given the uncertainties 18 and 22 mmHg.  You have not convinced me that this “confirms” a short-term impact.  Please clarify. 

L226-228: The fact that indoor concentrations of PAHs were lower during the Festival than before the Festival is interesting.  How much lower were the indoor concentrations? How much higher were the outdoor concentrations? Are the differences statistically significant? 

Minor Comments:

L12: migration ïƒ  movement

L15; to catch more concern on the impact ïƒ  examine the impact

L22-23 double check meaning

L25: had been ïƒ  have been

L25-30: These six lines contain two long sentences.  The second sentence repeats much of what is in the first sentence.  Perhaps use this for the second sentence: 

“In the future, studies focused on these processes are welcomed.”

L40: and also caused ïƒ  and it also causes

L46: Delete “Except for PMs,” Begin sentence with “Some gas …”

L50: the air quality ïƒ  poor air quality

L64: might also had ïƒ  might also have

L73: there are forbidden ïƒ  emissions are now forbidden

L81: critical review is available to fully investigate critical review that fully investigates

L107-108: ïƒ  Figure 1. The percent of studies focused on the Chinese Spring Festival by region. 

L127: found the WSOA ïƒ  found that the WSOA         

L134-136: Be careful with the rounding to be sure you don’t change the value by ~ a factor of 10. 

L143: cities had restricted ïƒ  cities that had restricted

L226: really scare ïƒ  really scarce. 

L251: the source, formation ïƒ  the source, information

L259: counted to reflex ïƒ  obtained to ascertain

L268: were given as below in detail ïƒ  are given in detail below

L277: there are part of studies ïƒ  there are some studies

L278: other studies ïƒ  additional studies

L280: more detail information ïƒ  more detailed information

L294: and then affected ïƒ  and then the pollution associated with fireworks affects

Reviewer 3 Report

Guixian et al., Presents the Chinese spring festival impact on air quality in China: A critical review, which is very interesting but unfortunately this study was not sufficiently improved after the first revision.

Analyzing the submitted work, the authors partially implemented the recommendations made (e.g. Fig. 1). Recommendations such as The presented results from previous research are not sufficient argued and comparatively treated. More comparative results should be extensively treated to evaluate the actual research stage’s pros and cons. I recommend that the authors be more specific and exclude affirmations like “some studies” without a clear reference and traceability were treated superficially.

The author does not respond to the comment.

Also for the comment: I strongly recommend adding more information about fireworks’ impact at different sites ( across the world).  Even though the study is focused on China, a subsection that should include the results from fireworks in other regions can be an added value to the paper.

The author does not respond to the comment and the answer for this is superficial.

 The recommendations of this study are still general without being interconnected with the existing needs on this topic.

Author Response

This manuscript is a resubmission of an earlier submission. The following is a list of the peer review reports and author responses from that submission.